# Next-Generation Sequencing Analysis of Pancreatic Cancer Using Residual Liquid Cytology Specimens from Endoscopic Ultrasound—Guided Fine-Needle Biopsy: A Prospective Comparative Study with Tissue Specimens

**DOI:** 10.3390/diagnostics13061078

**Published:** 2023-03-13

**Authors:** Hiromichi Iwaya, Akihide Tanimoto, Koshiro Toyodome, Issei Kojima, Makoto Hinokuchi, Shiroh Tanoue, Shinichi Hashimoto, Machiko Kawahira, Shiho Arima, Shuji Kanmura, Toshiaki Akahane, Michiyo Higashi, Shinsuke Suzuki, Shinichi Ueno, Takao Ohtsuka, Akio Ido

**Affiliations:** 1Digestive and Lifestyle Diseases, Graduate School of Medical and Dental Sciences, Kagoshima University, 8-35-1 Sakuragaoka, Kagoshima 890-8520, Japan; 2Iwaya Internal Medicine and Endoscopy Clinic, 1-16-3 Arata, Kagoshima 890-0054, Japan; 3Center for Human Genome and Gene Analysis, Graduate School of Medical and Dental Sciences, Kagoshima University, 8-35-1 Sakuragaoka, Kagoshima 890-8544, Japan; 4Department of Clinical Cancer Research, Graduate School of Medical and Dental Sciences, Kagoshima University, 8-35-1 Sakuragaoka, Kagoshima 890-8544, Japan; 5Department of Digestive Surgery, Breast and Thyroid Surgery, Graduate School of Medical and Dental Sciences, Kagoshima University, 8-35-1 Sakuragaoka, Kagoshima 890-8544, Japan

**Keywords:** cancer panel, endoscopic ultrasound–guided fine-needle aspiration, next-generation sequencing, pancreatic cancer

## Abstract

This study evaluated the feasibility and clinical utility of liquid-based cytology (LBC) specimens via endoscopic ultrasound–guided fine-needle biopsy (EUS-FNB) for next-generation sequencing (NGS) of pancreatic cancer (PC). We prospectively evaluated the performance of DNA extraction and NGS using EUS-FNB samples obtained from PC. Thirty-three consecutive patients with PC who underwent EUS-FNB at our hospital were enrolled. DNA samples were obtained from 96.8% of the patients. When stratified with a variant allele frequency (VAF) > 10% tumor burden, the NGS success rate was 76.7% (*n* = 23) in formalin-fixed paraffin-embedded (FFPE), 83.3% (*n* = 25) in LBC, and 76.7% (*n* = 23) in frozen samples. The overall NGS success rate was 86.7% (*n* = 26) using FFPE, LBC, or frozen samples. The detection rates for the main mutated genes were as follows: 86.7% for *KRAS*, 73.3% for *TP53*, 66.7% for *CDKN2A*, 36.7% for *SMAD4*, and 16.7% for *ARID1A*. LBC had the highest median value of VAF (23.5%) for *KRAS* and *TP53*. PC mutation analysis using NGS was successfully performed using LBC compared with FFPE and frozen samples. This approach provides an alternative and affordable source of molecular testing materials.

## 1. Introduction

The incidence of pancreatic cancer (PC) has increased in most high-income countries [1,2]. Surgical resection is the only curative treatment for PC; however, 70–80% of patients cannot undergo surgical resection and have a poor prognosis, with a 5-year overall survival rate of 8.2% [2,3,4,5]. Next-generation sequencing (NGS) is increasingly used to identify diagnostic, prognostic, and predictive mutations in many malignant tumors and has significantly improved treatment outcomes [6,7,8]. In PC, genome profiling data can improve patient treatment and survival [9,10]. Precision medicine that applies molecular targeted therapy based on gene expression and mutation profiling is expected to play a particularly important role in unresectable PC. The recent National Comprehensive Cancer Network (NCCN) guideline recommends germline testing for any patient with confirmed PC and gene profiling of tumor tissue, as clinically indicated for patients with locally advanced and metastatic disease [11]. In Japan, the OncoGuide^TM^ NCC Oncopanel System (NOP; Sysmex Corporation, Hyogo, Japan) and FoundationOne^®^ CDx (F1CDx; Foundation Medicine, Cambridge, MA, USA) are approved by insurance [12,13].

High-quality tumor samples are essential for comprehensive NGS analysis. To date, most genomic testing and biomarker research has been performed on specimens obtained from either endoscopic ultrasound–guided fine-needle aspiration (EUS-FNA)/fine-needle biopsy (FNB) or surgical resection in PC and solid tumors [14,15]. EUS-FNA/FNB is an established and highly accurate method used for PC diagnosis [16,17,18,19]. A recent study reported that EUS-FNB is a useful sampling technique for genomic profiling, DNA, and histology yield in patients with PC [20,21]. Hisada et al. showed that endoscopic ultrasound–guided tissue acquisition using a 19-G FNB needle is effective in achieving the proportion meeting the NOP analysis suitability criteria in patients with unresectable PC [22].

Genomic sequencing of PC is examined based on formalin-fixed paraffin-embedded (FFPE) tissues obtained from surgery and EUS-FNB. However, the FFPE process often has disadvantages, such as red blood cell contamination, deterioration of quality over time, and being prone to under-sampling for DNA extraction. Moreover, liquid-based cytology (LBC) is the most widely accepted technique for diagnosis of PC using EUS-FNA, which can better preserve DNA quality for NGS, as compared with FFPE, even after several years of storage [23,24,25,26,27,28]. Recent studies have reported that residual LBC tissue obtained from a tissue biopsy or surgical resection is a useful material for gene testing in patients with various cancers, such as lung cancer or gynecologic cancer [28,29,30]. However, to the best of our knowledge, studies on the utility of residual LBC for genomic profiling via NGS in patients with PC are scarce.

In this study, we established a small custom cancer gene panel with full exon coverage of 28 cancer-related genes and used this custom panel to comparatively analyze the genetic profiles obtained from residual LBC specimens and corresponding FFPE tissues with respect to the presence of somatic gene mutations. Therefore, this study aimed to evaluate the performance of gene profiling extracted from residual LBC specimens as compared with the corresponding FFPE and frozen specimens using EUS-FNB with a 22-gauge needle.

## 2. Materials and Methods

### 2.1. Patient Population and Study Design

This was a prospective, single-arm pilot study of the feasibility and efficacy of targeted NGS using EUS-FNB samples collected at one Japanese referral center. All consecutive patients aged ≥ 18 years initially diagnosed with PC on computed tomography (CT) or magnetic resonance imaging who required EUS-FNB were included in this study between November 2019 and March 2020 at Kagoshima University Hospital. FFPE, LBC, and frozen tissue samples were obtained with EUS-FNB. The exclusion criteria were as follows: (1) severe, poor medical condition for EUS-FNB; (2) no history of treatment, including chemotherapy or surgical resection; (3) benign case with EUS-FNB; (4) pregnancy; and (5) refusal to participate in this study (Figure 1).

The primary endpoint was to evaluate the sensitivity, specificity, and accuracy of the 22-gauge FNB needle for malignancy and to compare the adequacy of the quality of DNA extraction and gene testing among LBC, FFPE, and frozen samples. The secondary endpoint was to investigate the rate of mutated genes using targeted panel sequencing. 

### 2.2. EUS-FNB and Sampling Methods

Patients were placed under conscious sedation, and EUS-FNB was performed using a convex array echoendoscope (GF-UCT260; Olympus Optical Corp. Ltd., Tokyo, Japan, or EG-530UT2; Fujifilm Corp., Tokyo, Japan) and 22-gauge needles (Acquire; Boston Scientific, Marlborough, MA, USA). Three endoscopists with ≥15 years of experience in EUS determined the tumor target area and performed EUS-FNB. The needle was moved back and forth within the lesion at least 20 times with 10–20 cc of negative pressure and removed through the scope. Tumor tissues were dropped into formalin bottles using the stylet or air for histological evaluation; no rapid onsite evaluation was performed. The current guidelines recommend two to three needle passes with an FNB needle if ROSE is unavailable [31]. In a recent multicenter retrospective study, MOSE was reported to have a high diagnostic yield and accuracy and was associated with a large needle diameter and three or more needle passes [32]. Therefore, we conducted three to four needle passes, and white tissue was identified as a tumor sample to accurately select the tumor area.

Three tissue samples were obtained during one EUS-FNB session and used for FFPE, LBC, and frozen tissue processing. For diagnosis of PC, two FFPE specimens were stained with hematoxylin and eosin, while one specimen was prepared for a frozen sample. The residual LBC samples were collected after the FFPE process, and frozen samples were obtained. After the LBC samples were stored for 14 days at 4 °C in CytoRich Red solution, the cultured cells were fixed onto glass slides for Papanicolaou staining (Figure 2). Frozen samples were stored at −80 °C until DNA extraction. All specimens obtained with EUS-FNB were evaluated for malignancy during cytological and pathological examinations by more than two expert pathologists with ≥10 years of experience.

### 2.3. DNA Extraction and Quality Assessment

After EUS-FNB samples were pathologically identified as PC using hematoxylin-eosin (HE) and immunohistochemical staining with the MIB-1 labeling index, DNA was extracted from FFPE and LBC samples using the Maxwell 16 FFPE Tissue LEV DNA Purification Kit (Promega, Madison, WI, USA) and from frozen samples using the Monarch^®^ Genomic DNA Purification Kit (New England BioLabs Inc., Ipswich, MA, USA). DNA from whole blood (control; QIAamp DNA Blood Mini Kit; Qiagen, Hilden, Germany) and EUS-FNB samples was extracted to identify tumor-specific gene mutations. The DNA concentration was measured using the Qubit 3.0. Fluorometer dsDNA BR assay kit (Life Technologies, Carlsbad, CA, USA), and the DNA quality was confirmed using the QIAseq DNA Quantimize kit (Qiagen). Real-time PCR was performed using the QIAseq DNA Quantimize kit, and Ct values were calculated from 100 and 200 bp amplicons. The quality check (QC) score (ΔCt200 − ΔCt100/200 − 100) was calculated to evaluate the DNA quality, with lower QC scores indicating less DNA degradation. If the QC score was less than 0.004 and the amplification was more than 0.5, the DNA was considered to be of high quality.

### 2.4. Targeted Sequencing and Analysis

Samples used for NGS were subjected to pathological assessment, and areas including the tumor epithelium were identified via HE staining. The FFPE specimen available for NGS was 10 µm × 3, and the tumor percentage in the HE specimen was defined as 10% or more. Papanicolaou-stained specimens also had 10% or more tumor cells, and although cell counts were not performed, the pathologist and cytologist estimated the tumor cells per field of view according to previously reported methods and confirmed that there were sufficient numbers of cells. Twenty-eight cancer-related genes were selected from the QIAseq Targeted DNA Custom Panel (Qiagen, QIAseq DNA panel) comprising 1350 primers covering a total area of 98,555 bp of interest and an exon coverage of 100%. These genes were *KRAS*, *TP53*, *CDKN2A*, *SMAD4*, *EGFR*, *KMT2D*, *BRAF*, *CD79B*, *CD79A*, *ERBB2*, *CREBBP*, *NRAS*, *PIK3CA*, *GATA3*, *PTEN*, *FGFR1*, *FGFR2*, *FGFR3*, *RET*, *ARID1A*, *LRP1B*, *RB1*, *TERT*, *HRAS*, *CDH1*, *MYD88*, *ESR1*, and *CTNNB1*. Target genes were selected from the top five most frequently mutated genes in thyroid gland, breast, biliary tract, stomach, large intestine, liver, lung, endometrium, ovary, pancreas, and urinary tract cancers, as well as diffuse large B-cell lymphoma, according to the Catalog of Somatic Mutations in Cancer (COSMIC ver. 90, https://cancer.sanger.ac.uk/cosmic (accessed on 4 December 2019)). The TERT region includes untranslated regions and promoter regions. A total of 3–126 ng of DNA was diluted with hybridization buffer to a final concentration of 20 pM and used for QIAseq Targeted DNA Custom Panel library construction and applied to the MiSeq sequencer (Illumina, San Diego, CA, USA). Sequence data were analyzed using the Qiagen Web Portal service (https://www.qiagen.com/us/shop/genes-and-pathways/data-analysis-centeroverview-page/ (accessed on 4 December 2019)).

### 2.5. Statistical Analysis

The correlations between the variant allele frequency (VAF) from the FFPE process and LBC for the *KRAS* and *TP53* mutations were identified using Spearman correlation coefficients. All statistical analyses were conducted with IBM SPSS version 26.0 (IBM Corp., Armonk, NY, USA). A *p*-value of < 0.05 was considered significant.

## 3. Results

### 3.1. Patient Characteristics

The sensitivity, specificity, positive predictive value, negative predictive value, and accuracy of the 22-gauge FNB needle for 33 lesions with confirmed diagnosis were 96.9% (31/32), 100% (1/1), 100% (31/31), 50% (1/2), and 97% (32/33), respectively. The EUS-FNB samples obtained from 31 patients, excluding 2 patients with benign diagnosis and insufficient specimens, were included for DNA extraction and subsequent NGS. Patient characteristics are summarized in Table 1. Pathological diagnosis of EUS-FNB samples indicated that 30 of the 31 PC samples were pancreatic ductal adenocarcinoma (PDAC) and 1 was acinar cell carcinoma. Of the 31 patients with PC, 14 had resectable/borderline resectable disease, 7 had locally advanced disease, and 10 presented with metastatic disease. Lesions of metastasis were observed in nine liver, nine lymph node, and two peritoneum cancers. The lesion percentages in each Union for International Cancer Control (UICC) stage were 25.8% (*n* = 8) in stage I, 32.3% (*n* = 10) in stage II, 6.6% (*n* = 2) in stage III, and 35.5% (*n* = 11) in stage IV.

### 3.2. Quality of DNA Extracted from EUS-FNB Samples

The results of Qubit, amplifiable DNA, and QC scores are summarized in Table 2. The DNA was used for NGS (except for no. 8), and there were no samples with a QC score greater than 0.04. No significant DNA degradation was present to limit sample use for NGS analysis. Only one FFPE sample failed to yield DNA in no. 8, which was obtained from a tumor on the head side of the pancreas. The size of the tumor was 27 mm, and the UICC stage was II B. Specimens of no. 8 were insufficient tumor tissues for DNA extraction. Of the 31 samples, 30 (96.8%) had adequate DNA quantity, and these samples were successfully analyzed with FFPE (96.8%), LBC (100%), and frozen (100%) methods for NGS.

### 3.3. Gene Alteration Profiles

The detected gene mutation profiles are shown in Figure 3. Repeated detection of gene mutations included *KRAS* (86.7%), *TP53* (73.3%), *CDKN2A* (66.7%), *SMAD4* (36.7%), *ARID1A* (16.7%), *KMT2D* (6.7%), and *PTEN* (6.7%). The following genes accounted for less than 5% of the identified alterations: *PIK3CA*, *LRP1B*, *FGFR1*, and *ERBB2* (Figure 3). The results from NGS analysis of FFPE, LBC, and frozen samples are summarized in Table 3. LBC samples were most included at VAF > 10% and 30% except for VAF > 20%. The success rate of NGS with more than 10% VAF was 86.7% (*n* = 26) using FFPE, LBC, or frozen samples.

### 3.4. KRAS and TP53 VAF

*KRAS* and *TP53* mutations were most frequently detected in the 30 samples used for NGS analysis (Figure 3). The median rates of VAF in patients with *KRAS* and *TP53* were 17.43% and 21.96% in FFPE, 23.95% and 25.04% in LBC, 21.27% and 20.78% in frozen samples, respectively (Table 4). The LBC tissue specimens had the best VAF for targeted gene sequencing. The VAF from LBC samples was significantly correlated with that from FFPE samples for *KRAS* (r = 0.89, *p* < 0.0001) and *TP53* (r = 0.75, *p* < 0.0001); see Figure 4. Table 5 shows the frequency of VAF > 30% in patients with *KRAS* and *TP53* mutations. The rate of *KRAS* in samples with VAF > 30% from FFPE or frozen specimens was 27.3%, while that of *TP53* from LBC samples was 47.6%.

## 4. Discussion

This study prospectively investigated the quality of DNA extracted from samples isolated using EUS-FNB with a 22-guage needle and the feasibility of NGS analysis. EUS-FNB is a standard pathological examination that can promote precision medicine for patients with PC. Recently, a network meta-analysis showed that FNB needles (Franseen and Forktip needles) are significantly superior to reverse bevel and FNA needles [33]. Furthermore, regarding tissue collection methods in EUS-FNB, Facciorusso et al. compared the “slow-pull”, “dry suction”, “modified wet suction” (with the needle preflushed with 1–2 mL of saline after removing the stylet, application of suction using a 10 mL pre-vacuum syringe), and “no suction” techniques in a network meta-analysis [34]. They reported that the “wet suction” technique yields the most adequate samples. In this study, EUS-FNB was performed using the “dry suction” methods in all cases.

Several studies have demonstrated NOP analysis suitability criteria for EUS-FNB using a 19-guage needle for patients with unresectable PC [22,35]. Moreover, one prospective and two retrospective studies have reported on comprehensive genomic testing using 22-guage FNB needles [36,37,38]. Elhanafi et al. compared 22-guage FNA with 22-guage FNB needles for targeted NGS of PC. The authors reported that genomic testing could be performed in 70% of patients undergoing EUS tissue sampling with cytology diagnostic for PC and that the 22-guage FNB needle is more likely to yield samples sufficient for NGS as compared to the 22-guage FNA needle [37]. Moreover, Carrara et al. conducted a prospective comparative pilot study at a single-center hospital [36] that included 33 patients referred for locally advanced PC who underwent EUS-FNB using a 22-guage needle (AcquireTM). The authors demonstrated that tissue specimens obtained under EUS-FNB allowed for DNA sample extraction and subsequent NGS analysis in 97% of cases. Our analyses also revealed that DNA from PC samples obtained with EUS-FNB could be used for NGS to detect gene mutations with high probability. These results suggest that tissue sampling with EUS-FNB using a 22-guage needle is a suitable method for cancer genomic analysis, especially in patients in an advanced disease stage who cannot undergo curative resection. To the best of our knowledge, this is the first prospective study to investigate the feasibility of LBC in gene profiling using EUS-FNB with a 22-guage needle in patients with PC. In our study, the quality of DNA extracted from FFPE, LBC, and frozen samples was high regardless of the resectability and UICC stage of PC. For DNA extraction, microdissection of surgical specimens is generally recommended to ensure an adequate tumor burden [14]. However, our results suggest that EUS-FNB samples produce high-quality DNA even in cases of unresectable PC. Moreover, the total NGS success rate was 86.7% with VAF > 10%, which is comparable to previous reports using EUS-FNB samples [37,39] and resected specimens from solid tumors [14]. In targeted sequencing using an NGS oncogene panel, the sequence coverage of the target region is usually set to 250–500× or more, while the recommended VAF detection threshold is 5–10% [40], which indicates that even with EUS-FNB samples, gene detection can be performed with high probability at VAF >10% if the proper depth of sequence coverage is maintained. Gleeson et al. conducted a retrospective study of 29 patients with ampullary cancer or PC whose cytology smears were used for NGS with a panel of 160 cancer genes [15], and they showed that EUS-FNB specimens are similar to surgically resected specimens with respect to the accuracy in reflecting the tumor genomic profile. These findings indicate that targeted NGS cancer gene panels can be successfully generated using specimens from both surgical resection and EUS-FNB. EUS-FNB samples, therefore, can yield DNA with sufficiently high quality and quantities for NGS analysis, at least for relatively small panels. Whole-exome sequencing is not always necessary in targeted therapy, and these smaller panels may be sufficient to identify target genes to inform patient care.

Meanwhile, the risk of needle tract seeding after EUS-FNA should be considered, especially in body and tail lesions. A multicenter cohort study reported that EUS-FNA had no negative effects on patient survival, but needle tract seeding after EUS-FNA was observed to have a non-negligible rate in pancreatic body and tail cancer [41]. Since it remains unknown whether fine-needle aspiration procedure factors, such as the number of needle passes, are significantly associated with the occurrence of needle tract seeding, large-scale studies are warranted.

Although the quality of DNA extracted was comparable among FFPE (96.8%), LBC (100%), and frozen (100%) samples, more LBC (83.3%) samples successfully preserved VAF > 10% when compared with FFPE (76.6%) and frozen (76.7%) samples. Although the preparation and storage of FFPE tissues are recommended for appropriate NGS analyses [42], genomic DNA in FFPE and frozen tissues degrades over time and can result in insufficient amplification. Conversely, LBC samples reduce red blood cell contamination and increase the tissue collection efficacy of EUS-FNB sampling [23,27]. Moreover, they preserve the DNA quality and yield even after long-term storage and the absence of diagnostic material loss due to cell scraping [27].

However, the use of LBC specimens for NGS analysis has a few disadvantages. First, NGS analysis of LBC specimens is performed with pooled residual samples. Although we confirmed the presence of tumor cells in residual LBC specimens and subjected them to DNA extraction, false-negative materials may be applied to NGS. Second, LBC specimens may not always be suitable for NGS, because an efficient method to enrich the tumor cell fraction in LBC specimens has not been developed. An additional disadvantage is the labor- and time-intensiveness and the higher costs as compared to the conventional Papanicolaou smear. Therefore, improvement is required to develop the most appropriate gene profiling using residual LBC samples.

The frequency of mutations identified in this study is consistent with previous PC studies, with *KRAS*, *TP53*, *CDKN2A*, and *SMAD4* recorded as the most commonly mutated genes [5,9,35]. Using a customized gene panel of 28 cancer-related genes, genomic profiles were successfully generated from 26 of 30 patient samples (86.7%). Several tumors included mutations in genes that may be candidates for precision therapy, including mutations in *PTEN* (everolimus), *PIK3CA* (alpelisib), *FGFR1* (pazopanib, erdafitinib), and *ERBB2* (trastuzumab, pertuzumab).

NGS testing routinely identified clinically relevant mutations from 5 to 10% VAF of the recommended tumor burden for genomic analysis [43]. In our study, VAF from FFPE and LBC samples was comparable in the detection of *KRAS* and *TP53*. FFPE and LBC samples with VAF > 20% were both successful in identifying commonly mutated genes in PC. These results are consistent with previous reports showing that LBC samples obtained with EUS-FNB of pancreas, thyroid gland, and breast cancers are more useful for molecular testing when compared to FFPE samples [30,44]. This indicates that LBC samples are a complementary approach to the FFPE process, which is recommended for NGS analysis. The rate of tumor burden in the EUS-FNB sample was almost sufficient for use with a small cancer gene panel; however, the results were not entirely satisfactory, because a larger cancer panel or whole-genome sequencing required a resected tissue area of ≥5 mm^2^ or ≥20% of the total nucleated cells to be tumor cells [39,45,46]. The performance of targeted NGS depends on the sampling method used and is associated with tumor yield. Laboratory experience, specialized knowledge, and device development are required to further optimize the cellular density that can be collected from EUS-FNB specimens.

When FFPE samples were combined with LBC samples, the success rate of NGS improved compared to FFPE samples alone. Previous studies have shown that residual cytology extracted from EUS-FNA/FNB samples has sufficient feasibility and performance for NGS analysis [15,37,44,47]. Reynolds et al. showed that NGS analysis could be successfully performed on residual cell pellets obtained from LBC samples of lung adenocarcinoma and that their gene hotspot panel was able to identify clinically relevant EGFR variants, as well as mutations in *ERBB2*, *KRAS*, *MET*, and *PIK3CA* [44].

In this study, patients with or without surgical indication were enrolled. The NCCN guideline recommends germline testing for any patient with confirmed PC using comprehensive gene panels for hereditary cancer syndromes [11]. Post-surgical tissue has a risk of unsuccessful NGS caused by inadequate tumor yield since those with resectable and borderline resectable PC generally receive neoadjuvant chemotherapy. Therefore, gene analysis with preoperative EUS-FNA in any patient with PC is useful in gene profiling for recurrent PC.

This study had some limitations. First, selection bias might have occurred due to the study’s single-center nature and small patient population and the involvement of a small number of experienced endoscopists. Although the analysis did not demonstrate any difference in results based on the performing endoscopist, there was the potential risk of bias introduction by the operator. Thus, we prospectively analyzed the data of consecutive patients with PC at all stages of disease. Second, we used a small number genome profiling as an adequacy measurement. The 28 cancer-related genes did not include the *BRCA* gene, which is important in determining the treatment strategy for PC.

Although this is not comprehensive genomic profiling that is widely accepted as the standard of care for solid pancreatic masses, it provides the minimal tumor mutation profiling that can be used by physicians to make treatment decisions based on specific findings. This study investigated the success rate of NGS for not germline testing but gene profiling of tumor tissue. Our results reflect real-world clinical data and provide a guide for decision making in the treatment of patients with PC. Third, the cases in this study were only diagnosed by imaging and no final pathology diagnosis was performed on the resected specimens. Since the gene panel did not include GNAS, VHL, or RNF43, we could not rule out the possibility of IPMN-derived pancreatic cancer associated with these genes.

## 5. Conclusions

EUS-FNB samples can provide high-quality and sufficient amounts of DNA for NGS analysis, at least for relatively small gene panels. Furthermore, LBC specimens for NGS testing are an optional tool for genetic testing as a diagnostic or therapeutic strategy in patients with any PC.

## Figures and Tables

**Figure 1 diagnostics-13-01078-f001:**
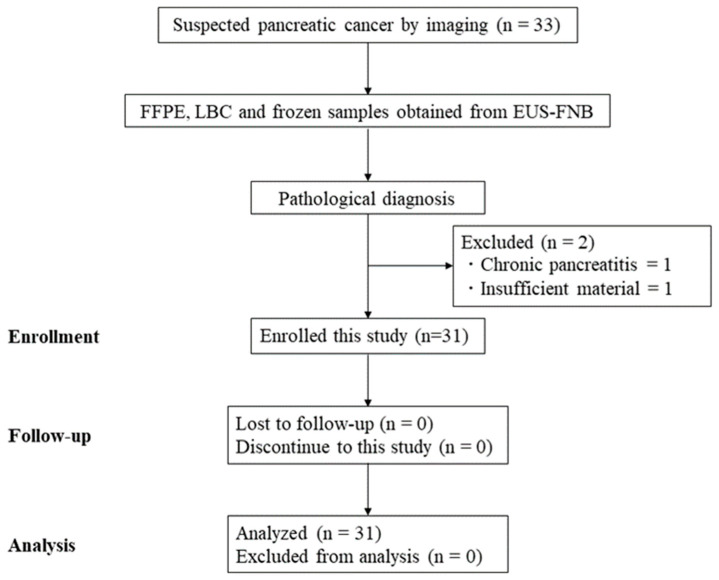
Flow diagram of patient selection. CT, computed tomography; FFPE, formalin-fixed paraffin-embedded; LBC, liquid-based cytology; EUS-FNB, endoscopic ultrasound–guided fine-needle biopsy.

**Figure 2 diagnostics-13-01078-f002:**
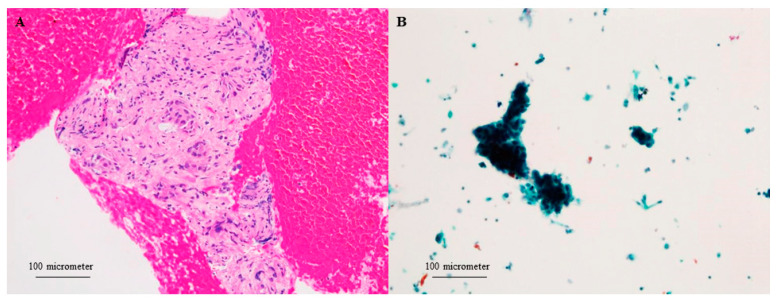
Scanned magnification (200×) image of hematoxylin-eosin staining (**A**) and liquid-based cytology (**B**) using a 22-gauge needle for pancreatic adenocarcinoma.

**Figure 3 diagnostics-13-01078-f003:**
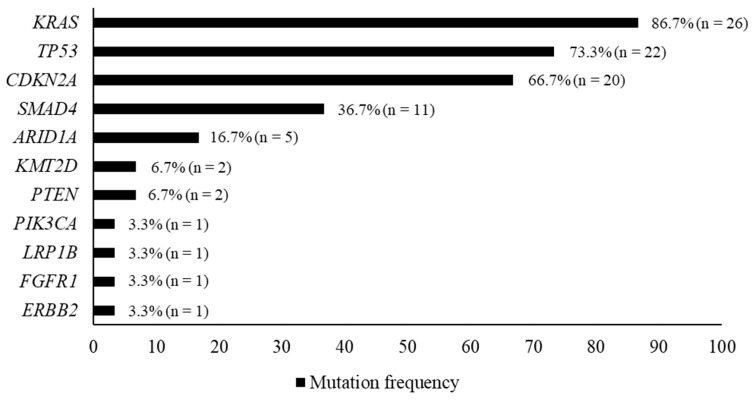
Frequency of mutations identified with genomic testing of 30 patients.

**Figure 4 diagnostics-13-01078-f004:**
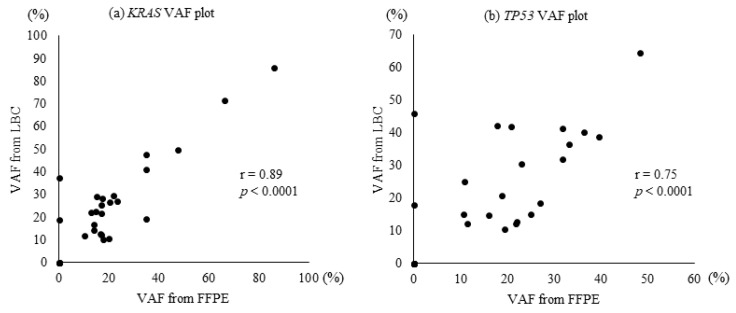
Correlations of VAF from LBC with FFPE specimens in the plots of *KRAS* and *TP53* gene expression. VAF, variant allele frequency; LBC, liquid-based cytology; FFPE, formalin-fixed paraffin-embedded.

**Table 1 diagnostics-13-01078-t001:** Patient characteristics.

	Patients (*n* = 31)
Sex, *n* (%)	
Male	18 (58.1)
Female	13 (41.9)
Age, years, median (range)	69 (36–89)
Location, *n* (%)	
Head	16 (51.6)
Body and tail	15 (48.4)
Pathological diagnosis, *n* (%)	
Pancreatic ductal adenocarcinoma	30 (96.8)
Acinar cell carcinoma	1 (3.2)
Tumor size, mm, median (range)	28.5 (15–61)
Resectability classification, *n* (%)	
Resectable	11 (35.5)
Borderline resectable	3 (9.7)
Unresectable, locally advanced	7 (22.6)
Distant metastasis	10 (32.2)
Lesion of metastasis, *n* (%)	14 (45.2)
Liver	9 (29)
Lymph node	9 (29)
Peritoneum	2 (6.5)
Tumor marker level	
Carcinoembryonic antigen, median (range), ng/mL	2.8 (1.2–176)
Carbohydrate antigen 19-9, median (range), U/mL	203.6 (1.5–36,898)
Union for International Cancer Control stage, *n* (%)	
I	8 (25.8)
II	10 (32.3)
III	2 (6.6)
Ⅳ	35.5

**Table 2 diagnostics-13-01078-t002:** Amplifiable DNA and quality check scores in all patients with PC.

	Amplifiable DNA (ng/µL)			QC Score		
Patient	FFPE	LBC	Frozen	FFPE	LBC	Frozen
1	8.756	5.651	11.661	0.004	0.004	0.002
2	10.943	17.171	5	0.005	0.003	0.001
3	8.706	28.679	13.287	0.004	0.004	0.001
4	11.29	25.082	12.628	0.008	0.001	0.002
5	14.29	15.298	8.796	0.006	0.002	0.004
6	17.654	22.064	7.855	0.004	0.006	0.006
7	5.093	14.489	6.484	0.006	0.001	0
8	0.063	22.947	5.029	0.152	0.002	0
9	8.971	4.348	10.718	0.006	0.007	0.001
10	12.044	17.921	11.29	0.006	0.006	0.003
11	13.426	16.04	12.998	0.005	0.001	0.002
12	11.251	15.333	13.503	0.008	0.003	0.002
13	3.568	2.959	14.983	0.048	0.006	−0.001
14	18.618	6.659	15.263	0.002	0.002	−0.001
15	3.05	5.224	9.783	0.011	0.006	0.001
16	8	4	10.425	0.008	0.008	0.005
17	8.645	14.845	7.535	0.008	0.005	0.003
18	7.228	17.151	10.012	0.006	0.003	0
19	6.514	15.819	10.842	0.011	0.008	0.005
20	12.585	9.526	10.449	0.007	0.003	0.001
21	11.303	17.512	10.656	0.005	0.002	−0.001
22	5.135	10.956	9.908	0.007	0.002	0.001
23	6.582	13.272	10.401	0.008	0.004	0.003
24	14.126	25.315	12.226	0.007	0.005	0.003
25	4.954	6.807	12.716	0.011	0.005	0.002
26	4.59	10.83	13.195	0.011	0.008	0.005
27	7.261	15.07	6.697	0.006	0.001	0
28	14.709	12.178	11.865	0.008	0.006	0.003
29	8.448	12.687	11.906	0.007	0.002	0.002
30	8.576	13.287	11.554	0.008	0.004	0.003
31	7.693	15.674	11.225	0.01	0.003	0

**Table 3 diagnostics-13-01078-t003:** VAF rate of each sample in successful NGS.

		Rate of VAF (*n* = 30)	
	>10%	>20%	>30%
FFPE	76.7% (*n* = 23)	50.0% (*n* = 15)	26.7% (*n* = 8)
LBC	83.3% (*n* = 25)	50.0% (*n* = 15)	36.7% (*n* = 11)
Frozen	76.7% (*n* = 23)	53.3% (*n* = 16)	23.3% (*n* = 7)
FFPE or LBC or frozen	86.7% (*n* = 26)	70.0% (*n* = 21)	43.3% (*n* = 13)

**Table 4 diagnostics-13-01078-t004:** VAF values in patients with *KRAS* and *TP53* among FFPE, LBC, and frozen samples.

	*KRAS*		*TP53*	
	Median	Range	Median	Range
FFPE	17.43%	10.12–85.86%	21.96%	10.64–48.3%
LBC	23.95%	10.16–85.78%	25.04%	10.58–64.49%
Frozen	21.27%	10.38–81.61%	20.78%	8.26–45.35%

**Table 5 diagnostics-13-01078-t005:** Frequency of VAF > 30% in patients with *KRAS* and *TP53* mutations.

	VAF > 30%	
	*KRAS*	*TP53*
FFPE	27.3% (6/22)	31.6% (6/19)
LBC	25% (6/24)	47.6% (10/21)
Frozen	27.3% (6/22)	15% (3/20)

## Data Availability

The data presented in this study are available upon request from the corresponding author.

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
