# Peer review of "Next-Generation Sequencing Analysis of Pancreatic Cancer Using Residual Liquid Cytology Specimens from Endoscopic Ultrasound—Guided Fine-Needle Biopsy: A Prospective Comparative Study with Tissue Specimens"

_diagnostics, 2023, doi:10.3390/diagnostics13061078_

Round 1

Reviewer 1 Report

Very interesting prospective study. I have just some concerns on the novelty of authors' findings that have already been reported in previous studies (acknowledged by the authors themselves in the bibliography), for example the study by Carrara S et al. So the authors should point out further the novelty of their study. Other comments:

1) In the title the author use the term "cytology" but the study was conducted on histological samples collected with FNB needles.

2) Please report adequacy, accuracy, sensitivity and specificity rates

3) The authors should comment on the state of the art in the field, citing some recent papers on these new end-cutting needles (cite PMID: 35124072 and PMID: 36657607)

4) Was ROSE available in this study? Or was MOSE practiced? please comment

Author Response

Comments to the editor

We have modified the manuscript to adhere to the word limit for original articles. These changes did not affect the results or conclusions of our report. We sincerely thank you and the reviewers for the critical review of our manuscript. We have addressed the issues raised below and think that the revised manuscript has substantially improved.

Response to Reviewer

We would like to thank the reviewers for taking the time and effort to read our manuscript and providing meaningful comments/criticism that we have used to improve the quality of the manuscript.

Reviewer #1

Very interesting prospective study. I have just some concerns on the novelty of authors' findings that have already been reported in previous studies (acknowledged by the authors themselves in the bibliography), for example the study by Carrara S et al. So the authors should point out further the novelty of their study. Other comments:

Response: We thank the reviewer for the meaningful comment. We apologize for not sufficiently pointing out the novelty of this study in the manuscript. We have modified the Discussion section.

Please also refer to our response to comment R2-1 below.

[1st comment of Reviewer 1 (R1-1)] In the title the author use the term "cytology" but the study was conducted on histological samples collected with FNB needles.

Response: We thank the reviewer for the comment. We apologize for any confusion caused by the inconsistency in the title. We have changed the title as below.

Original title: A prospective analysis of next-generation sequencing for pancreatic cancer using liquid-based cytology specimens by endoscopic ultrasound-guided fine-needle biopsy

New title: Next-generation sequencing analysis of pancreatic cancer using residual liquid cytology specimens from ultrasound endoscopy-guided fine-needle biopsy: a prospective comparative study with tissue specimens

[2nd comment of Reviewer 1 (R1-2)] Please report adequacy, accuracy, sensitivity and specificity rates

Response: We thank the reviewer for the thoughtful comment. We apologize for the insufficient explanation regarding the results of EUS-FNB. We have changed the following text in the "Materials and Methods" and "Results" sections.

In Materials and Methods

Original sentences: The primary endpoint was to compare the adequacy for the quality of DNA extrac-tion and gene testing among LBC and FFPE, frozen samples.

New sentences (page 2): The primary endpoint was to evaluate the sensitivity, specificity, and accuracy of the 22G FNB needle for malignancy and to compare the adequacy of the quality of DNA ex-traction and gene testing among LBC, FFPE, and frozen samples.

In Results

Original sentences: The EUS-FNB samples obtained from 31 patients were included in the analysis of DNA extraction and subsequent NGS.

New sentences (page 5): The sensitivity, specificity, positive predictive value, negative predictive value, and accuracy of the 22G FNB needle for 33 lesions with confirmed diagnosis were 96.9% (31/32), 100% (1/1), 100% (31/31), 50% (1/2), and 97% (32/33), respectively. The EUS-FNB samples obtained from 31 patients, excluding 2 patients with benign diagnosis and insufficient specimens, were included for DNA extraction and subsequent NGS.

[3rd comment of Reviewer 1 (R1-3)] The authors should comment on the state of the art in the field, citing some recent papers on these new end-cutting needles (cite PMID: 35124072 and PMID: 36657607)

Response: We thank the reviewer for the meaningful comment. We apologize for the lack of knowledge on the new FNB needle. We have cited the paper the reviewer indicated and described the usefulness of EUS-FNB with new needles and the appropriate tissue collection methods in the Discussion.

Original sentences: This study prospectively investigated the quality of DNA extracted from samples isolated using EUS-FNB with 22-G needle and the feasibility of NGS analysis. EUS-FNB is a standard pathological examination that can promote precision medicine for patients with PC.

New sentences (page 8): This study prospectively investigated the quality of DNA extracted from samples isolated using EUS-FNB with a 22-G needle and the feasibility of NGS analysis. EUS-FNB is a standard pathological examination that can promote precision medicine for patients with PC. Recently, a network meta-analysis showed that FNB needles (Franseen and Forktip needles) were significantly superior to reverse bevel and FNA needles [33]. Furthermore, regarding tissue collection methods in EUS-FNB, Facciorusso et al. compared the "slow-pull", "dry suction", "modified wet suction"(with the needle preflushed with 1–2 mL of saline after removing the stylet, application of suction using a 10-mL pre-vacuum syringe), or "no suction" techniques in a network meta-analysis [34]. They reported that the "wet suction" technique yielded the most adequate samples. In this study, EUS-FNB was performed using the "dry suction" methods in all cases.

[4th comment of Reviewer 1 (R1-4)] Was ROSE available in this study? Or was MOSE practiced? please comment

Response: We thank the reviewer for the meaningful comment. We have changed and inserted the following sentences in the Materials and Methods.

Original sentences: Tumor tissues were dropped into formalin bottles using the stylet or air for histological evaluation; no rapid onsite evaluation was performed. Therefore, white tissue was identified as a tumor sample to accurately select the tumor area.

New sentences (page 3): Tumor tissues were dropped into formalin bottles using the stylet or air for histological evaluation; no rapid onsite evaluation was performed. The current guidelines recommend two to three needle passes with an FNB needle if ROSE is unavailable [31]. In a recent multicenter retrospective study, MOSE was reported to have a high diagnostic yield and accuracy and was associated with a large needle diameter and three or more needle passes [32]. Therefore, we conducted three to four needle passes and white tissue was identified as a tumor sample to accurately select the tumor area.

Reviewer #2

[1st comment of Reviewer 2 (R2-1)] Why did you choose a 22G needle instead of an 19G?

Response: We thank the reviewer for the helpful comments. We agree that FNB with the 19G needle is effective in combination with endoscopic ultrasound-guided tissue acquisition (EUS-TA) for genomic profiling. A reason for using a 22G needle in this study is that EUS-TA with such a needle is more commonly used than that with a 19G needle in real-world clinical practice for all stages of pancreatic cancer. Moreover, puncture with a 19G needle is often difficult for some lesion sites, such as uncinate lesions. Another reason is that there are still few reports on comprehensive genomic testing using 22G needles. We have changed or inserted the following sentences in the Discussion.

Original sentence: Several studies have demonstrated NOP analysis suitability criteria for EUS-FNB us-ing a 19-G needle for patients with unresectable PC [22, 31]. Moreover, two retrospective and one prospective studies have reported on comprehensive genomic testing using 22-G FNB needle [32-34]. Elhanafi et al. compared a 22-G FNA with 22-G FNB needles for tar-geted NGS of PC. The authors reported that genomic testing can be performed in 70% of patients undergoing EUS tissue sampling with cytology diagnostic for PC and that 22-G FNB needle is more likely to yield samples sufficient for NGS as compared with 22-G FNA needle [33]. Meanwhile, Carrara et al. conducted a prospective comparative pilot study at single-center hospital [34] that included 33 patients referred for locally advanced PC who underwent EUS-FNB using a 22-G needle (AcquireTM). The authors demonstrated that tis-sue specimens obtained under EUS-FNB allowed for DNA sample extraction and subse-quent NGS analysis in 97% of cases. Our analyses also revealed that DNA from PC sam-ples obtained from EUS-FNB could be used for NGS to detect gene mutations with high probability. These results suggested that tissue sampling with EUS-FNB using a 22-G needle is a successful method for cancer genomic analysis, especially in patients with advanced disease who cannot undergo curative resection. To the best of our knowledge, this is the first prospective study to investigate the feasibility of LBC in gene profiling us-ing EUS-FNB with 22-G needle in PC patients.

New sentence (Page 9): Several studies have demonstrated NOP analysis suitability criteria for EUS-FNB us-ing a 19-G needle for patients with unresectable PC [22, 35]. Moreover, two retrospective and one prospective studies have reported on comprehensive genomic testing using 22-G FNB needles [36-38]. Elhanafi et al. compared 22-G FNA with 22-G FNB needles for tar-geted NGS of PC. The authors reported that genomic testing could be performed in 70% of patients undergoing EUS tissue sampling with cytology diagnostic for PC and that the 22-G FNB needle is more likely to yield samples sufficient for NGS as compared with the 22-G FNA needle [37]. Moreover, Carrara et al. conducted a prospective comparative pilot study at single-center hospital [36] that included 33 patients referred for locally advanced PC who underwent EUS-FNB using a 22-G needle (AcquireTM). The authors demonstrated that tissue specimens obtained under EUS-FNB allowed for DNA sample extraction and subsequent NGS analysis in 97% of cases. Our analyses also revealed that DNA from PC samples obtained from EUS-FNB could be used for NGS to detect gene mutations with high probability. These results suggest that tissue sampling with EUS-FNB using a 22-G needle is a suitable method for cancer genomic analysis, especially in patients in an ad-vanced disease stage who cannot undergo curative resection. To the best of our knowledge, this is the first prospective study to investigate the feasibility of LBC in gene profiling using EUS-FNB with a 22-G needle in PC patients.

[2nd comment of Reviewer 2 (R2-2)] How many tissue blocks with how many tumor cells were available for NGS?

Response: We thank the reviewer for the thoughtful comment. We apologize for the inadequate explanation of the NGS testing procedure. The FFPE was 10 µm × 3, and the tumor percentage was greater than 10% in the HE specimen. Papanicolaou-stained specimens also contained 10% or more tumor cells, and although cell counts were not performed, the pathologist and cytologist estimated the number of tumor cells per field of view according to previously reported methods and confirmed that there were sufficient cells. We inserted the following sentences in the Materials and Methods.

Original sentence: Samples used for NGS were subjected to pathological assessment and areas includ-ing the tumor epithelium were identified via HE staining.

New sentence (Page 4): Samples used for NGS were subjected to pathological assessment, and areas including the tumor epithelium were identified via HE staining. The FFPE available for NGS was 10 µm × 3, and the tumor percentage in the HE specimen was defined as 10% or more. Papanicolaou-stained specimens also had 10% or more tumor cells, and although cell counts were not performed, the pathologist and cytologist estimated the tumor cells per field of view according to previously reported methods and confirmed that there were sufficient numbers of cells.

[3rd comment of Reviewer 2 (R2-3)] Did you investigate GNAS, VHL and RNF43? You only had imaging, and it might be useful to rule out IPNM. There, GNAs could be useful.

Response: We thank the reviewer for the meaningful comments. We agree that the fact that the final pathology diagnosis was based on imaging alone and not on resection and that we could thus not rule out IPMN is a very important limitation. We inserted the following as a limitation in the Discussion.

Original sentence: Our results reflect real-world clinical data and provide a guide for decision making during the treatment of patients with PC.

New sentence (Page 11): Our results reflect real-world clinical data and provide guidelines for decision making in the treatment of patients with PC. Third, the cases in this study were only diagnosed by imaging, and no final pathology diagnosis was performed on the resected specimens. Since the gene panel did not include GNAS, VHL, or RNF43, we could not rule out the possibility of IPMN-derived pancreatic cancer associated with these genes.

 [4th comment of Reviewer 2 (R2-4)] Could you describe the mutations further?

Response: Thank you for your meaningful comments. We apologize for the inadequate explanation of the details of the mutations. Please refer to the attached VCF files of all analyzed cases for mutations.

References

  1. Polkowski, M.; Jenssen, C.; Kaye, P.; Carrara, S.; Deprez, P.; Gines, A.; Fernández-Esparrach, G.; Eisendrath, P.; Aithal, G.P.; Arcidiacono, P.; et al. Technical aspects of endoscopic ultrasound (EUS)-guided sampling in gastroenterology: European Society of Gastrointestinal Endoscopy (ESGE) Technical Guideline - March 2017. Endoscopy. 2017, 49, 989-1006.
  2. Mangiavillano, B.; Frazzoni, L.; Togliani, T.; Fabbri, C.; Tarantino, I.; De Luca, L.; Staiano, T.; Binda, C.; Signoretti, M.; Eusebi, L.H.; et al. Macroscopic on-site evaluation (MOSE) of specimens from solid lesions acquired during EUS-FNB: multicenter study and comparison between needle gauges. Endosc Int Open. 2021, 9, E901-e906.
  3. Gkolfakis, P.; Crinò, S.F.; Tziatzios, G.; Ramai, D.; Papaefthymiou, A.; Papanikolaou, I.S.; Triantafyllou, K.; Arvanitakis, M.; Lisotti, A.; Fusaroli, P.; et al. Comparative diagnostic performance of end-cutting fine-needle biopsy needles for EUS tissue sampling of solid pancreatic masses: a network meta-analysis. Gastrointest Endosc. 2022, 95, 1067-1077.e15.
  4. Facciorusso, A.; Crinò, S.F.; Ramai, D.; Madhu, D.; Fugazza, A.; Carrara, S.; Spadaccini, M.; Mangiavillano, B.; Gkolfakis, P.; Mohan, B.P.; et al. Comparative Diagnostic Performance of Different Techniques for Endoscopic Ultrasound-Guided Fine-Needle Biopsy of Solid Pancreatic Masses: A Network Meta-analysis. Gastrointest Endosc. 2023.

Reviewer 2 Report

Thank you for an interesting manuscript. 

1. Why did you choose a 22G needle instead of an 19G?

2. How many tissue blocks with how many tumor cells were available for NGS?

3. Did you investigate GNAS, VHL and RNF43? You only had imaging, and it might be useful to rule out IPNM. There, GNAs could be useful. 

4. Could you describe the mutations further?

4. 

Author Response

(The authors gave the same response as above.)
